# Hematological Changes in Sika Doe and Suckling Fawn Fed with Spent Mushroom Substrate of *Pleurotus ostreatus*

**DOI:** 10.3390/ani12151984

**Published:** 2022-08-05

**Authors:** Chongshan Yuan, Changze Li, Xinyuan Chen, Syed Muhammad Tahir, Aiwu Zhang, Min Wu

**Affiliations:** College of Animal Science and Technology, Jilin Agricultural University, Changchun 130118, China

**Keywords:** sika deer, digestibility, blood hematological, spent mushroom substrate

## Abstract

**Simple Summary:**

Sika deer velvet antler is a highly valued nutraceutic in traditional Chinese medicine with extremely high market value. Therefore, it is very important to find low-priced feed and reduce the feeding cost of sika deer to improve profitability. Mushrooms can be grown on agricultural wastes such as straw, cottonseed hull, and corn stalks, but with the process of cultivating mushrooms, spent mushroom substrate (SMS) has also been produced. Improper handling of SMS could still cause environmental pollution. The results show that SMS of *Pleurotus ostreatus* (SMS-MP) can be digested by sika doe and has no adverse effect on suckling fawn. At the same time, SMS-MP reduces the feeding cost of sika doe and avoids the pollution caused by SMS-MP to the environment.

**Abstract:**

Sika deer velvet antler is the most important animal nutraceutic in traditional Chinese medicine. Reducing the breeding cost of sika deer by looking for a low-cost diet is the main research direction at present. The purpose of this experiment was to find an alternative diet for sika deer and reduce the cost of the diet by using spent mushroom substrate (SMS) as a concentrate supplement. The apparent digestibility for sika doe and the hematological changes of sika doe and suckling fawn were measured by replacing 10% of the concentrate supplement with SMS of *Pleurotus ostreatus* (SMS-MP). Compared with the control group, the digestibility of dry matter (DM), total protein (TP), globulin (GLO), and cholesterol (CHOL) of sika doe were significantly decreased (*p* < 0.05), and glucose (GLU), alanine (Ala), phenylalanine (Phe), and proline (Pro) of sika doe were significantly increased (*p* < 0.05) after the replacement of SMS-MP. Compared with the control group, the serum GLU of suckling fawn was significantly decreased (*p* < 0.05) and the phosphatase (ALP) was significantly increased after the replacement of SMS-MP (*p* < 0.05). There were no significant differences in the immune globulin and amino acid of suckling fawns between the two groups (*p* > 0.05). The present findings confirm the applicability of SMS-MP as a sika doe concentrate supplement. At the same time, using SMS, a waste resource, can not only reduce the breeding cost of sika doe, but also make full use of SMS to reduce environmental pollution.

## 1. Introduction

Various agricultural residues such as corncob, straw, and cottonseed hull can be used as a medium for mushrooms, and the medium remaining after mushroom collection is called spent mushroom substrate (SMS) [1]. Although the use of agricultural residues has reduced environmental pollution to a certain extent, with the increasing demand for mushrooms [2], the remaining SMS after mushroom collection has become a serious source of pollution [3]. The lack of effective treatment methods for the large amount of SMS generated by the mushroom industry has caused serious pollution to the environment. Nowadays, many studies have been performed on how to make full use of SMS as a waste resource. It was reported that co-composting of SMS with pig manure accelerates the conversion of lignocellulose [4], and in addition, SMS and pig manure composting improves the growth of tomato and pepper seedlings compared with the use of commercial peat [5]. Another study showed that SMS could increase soil organic matter and its nutrient content, which was safer for soil than chemical fertilizers [6]. *Pleurotus ostreatus* is one of the common edible fungi in daily life. The spent mushroom substrate of *Pleurotus ostreatus* (SMS-MP) can be used in nurseries [7], bioremediation [8], and biofuel [9]. Another study found that SMS-MP was found to be a versatile, low-cost organic substrate that could reduce polychlorinated biphenyl (PCB) contamination in soil by activating the oxidation process of highly chlorinated PCBs [10]. Although SMS has been fully utilized at present, the current accumulation of SMS in the main mushroom-producing areas of China is alarming and has caused serious environmental pollution. Therefore, there is an urgent need to expand the potential use of SMS.

In addition, many studies have been conducted in recent years on the processing methods and advantages of SMS alternative feeds [11,12]. SMS can also be used as an animal feed supplement and can be easily digested by ruminants, providing additional animal feed resources [13]. It has been reported that SMS can be used as a functional feed additive to effectively improve milk production and hematological parameters in dairy cows [14]. However, there is no relevant study of SMS on the hematology of sika doe and suckling fawn ruminants. We speculate that sika deer can also digest SMS well and that it has no adverse effects on the hematology of sika doe and suckling fawn. Sika deer velvet antler is a valuable nutraceutic in traditional Chinese medicine and is widely recognized in many countries of the world [15]. Therefore, reducing the cost of raising sika deer is an important research direction to expand the deer industry. Our previous study has shown that 10% of SMS-MP can be safely replaced with a concentrated supplement in male sika deer with no adverse effects on nutrient digestion and hematology [16]. The purpose of this study was to evaluate the effectiveness of SMS-MP replacing 10% concentrate supplementation on the hematology and digestion in sika doe and the hematology of suckling fawn, reducing feeding costs while avoiding the environmental pollution of SMS-MP.

## 2. Materials and Methods

### 2.1. Experimental Designs

The experiment was carried out within 3 months of the lactation period of sika does, and 16 sika does with the same age and physiological condition were selected and divided into 2 groups randomly (8 animals/group). Sixteen suckling fawns were the babies of sika does. All sika deer lived in the artificial feedlot without weeds, shrubs, trees, etc. According to our previous study [16], the control group was fed normally and the experimental group was fed with corncob SMS-MP (dried), replacing 10% of the concentrate supplement. The diets were restricted and offered 3 times per day at 4:30~5:30, 10:30~11:30, and 16:30~17:30. Fresh, clean, drinking water and silage were available to the deer at all times. The nutrient levels of SMS-MP were shown in our previous study [17]. The ingredients of the concentrate supplement are shown in Table 1.

### 2.2. Sample Treatment

#### 2.2.1. Feeds and Feces Sampling

In the last 4 days of the trail, the samples of fresh silage, concentrate supplements, and feces were collected daily when the sika deers’ foraging finished. The samples were dried at a temperature of 65 °C for at least 48 h, then pulverized and stored in ziplock bags. Feed intake was calculated daily by weighing offered silage and concentrate supplement and rejections from the previous day.

#### 2.2.2. Blood Sampling

In the morning at the end of the experiment (before feeding), sika does and suckling fawns were anesthetized with xylazine hydrochloride (Hanhe Animal and Plant Medicine Co., Ltd., Qingdao, China) at a dose of 2.0 mL/100 Kg body weight. We used a disposable needle and a syringe to draw 10 mL of blood from the jugular vein, 5 mL of which was transferred to whole blood vacuum tubes and placed at 4 °C for 30 min to separate serum. An additional 5 mL of blood was transferred to vacuum tubes containing EDTA (Vacutainer; Becton Dickinson and Co. Rutherford, NJ, USA), and the vacuum tube was gently rocked to mix the EDTA with the blood. Immediately after collection, a centrifuge was used (3000 rpm for 15 min at 22 °C) to separate plasma. Then, serum and plasma samples were transferred to clean plastic tubes (2 mL each; Eppendorf, Hamburg, Germany). Samples were stored at −80 °C for later analysis. The plasma was used to detect the amino acids, while the serum was used to detect the biochemical parameters and immune globulin.

### 2.3. Determination

#### 2.3.1. Nutrient and Digestibility

Dry matter (DM): The samples were dried at 105 °C and cooled quickly to room temperature in a desiccator containing copper sulfate after constant weight, then determined by calculating the difference in sample weight.

Crude protein (CP): The LECO FP-528N/Protein Tester (LECO, Corporation, St. Joseph, MI, USA) was used to determine the CP content of samples according to the manufacturer’s instructions.

Ether extract (EE): The petroleum ether and the sample were put, tightly wrapped with filter paper, in the Soxhlet degreaser and extract at 70 °C for at least 16 h. Finally, the EE content was detected by calculating the difference in weight of the samples before and after extraction.

Organic matter (OM): The samples were put into the muffle furnace at 525 °C and burned for at least 12 h, and the content of OM was calculated by weighing the crude ash in the samples.

Energy: Energy was measured by using bomb calorimetry (Model 6050, Parr Instrument Company, Moline, IL, USA) according to the manufacturer’s instructions.

Calcium (Ca): After the sample was digested with concentrated sulfuric acid, 20 mL of the sample solution was transferred to a beaker to adjust the pH value. Then, 10 mL of saturated ammonium oxalate solution was added. The solution was boiled to obtain calcium oxalate precipitate, and the Ca content was determined by titration with potassium permanganate solution.

Phosphorus (P): After the sample was digested with concentrated sulfuric acid, the sample solution was mixed with ammonium vanadium molybdate chromogenic reagent, placed at room temperature for more than 10 min, and the absorbance of each solution was measured with a spectrophotometer at a wavelength of 400 nm. Then, a calculation of the P content of the sample from the standard curve was made.

Nutrient digestibility: Using acid-insoluble ash as an internal digestibility marker, the apparent digestibility was calculated using the following equation [18]:(1)$$N=100-100\times \frac{D}{F}\times \frac{D_{1}}{F_{1}}$$

*N* is the apparent digestibility; *D* is the indicator content in the diet; *F* is the indicator content in the feces; *F*_1_ is the nutrient content in the feces; *D*_1_ is the nutrient content in the diet.

#### 2.3.2. Analysis of Blood

Plasma-free amino acid: Plasma-free amino acid was determined using a high-performance liquid chromatography analyzer (Shimadzu Ltd., Kyoto, Japan). The experimental operation details are as described in the previous study [19].

Serum biochemical parameters: The concentration of serum biochemical parameters was analyzed according to the manufacturer’s instructions of BC-5300vet automatic blood cell analyzer (Biosystems SA, Barcelona, Spain).

Serum immune globulin: An enzyme-linked immunoassay kit (Enzyme Biotechnology Co., Ltd., Shanghai, China) was used to detect immune globulin of serum. Briefly, samples, biotin-labeled recognition antigen, and avidin-HRP were added to the enzyme-labeled well, respectively, incubated at 37 °C for 1 h and washed 5 times with phosphate buffer solution (PBS). Then, solution A, solution B, and stop solution were added, respectively, incubated at 37 °C for 10 min, and the values measured at 450 nm when the color changed from blue to yellow.

### 2.4. Statistical Analysis

All data are presented as least squares means ± SD. Comparison of significance was carried out by *t*-test, following the general linear model’s procedure of SPSS (SPSS 19.0; SPSS Inc., Chicago, IL, USA). Trends were considered significant when *p* < 0.05 was calculated. Trends were considered highly significant when *p* < 0.01 was calculated.

## 3. Results

### 3.1. Nutritional Ingredients of Concentrate Supplement

As shown in Table 2, there were no significant differences in nutritional ingredients of SMS-MP among the groups (*p* > 0.05), indicating that 10% SMS-MP did not affect the nutritional content of the doe concentrate supplement.

### 3.2. Feed Intake and Nutrient Digestibility of Sika Doe

There was no significant difference in feed intake and apparent digestibility between the two groups (*p* > 0.05), except that the digestibility of DM in the control group was significantly higher than that in the experimental group (*p* < 0.05), as indicated in Table 3.

### 3.3. Effects on Serum Biochemical Parameters

The effects of the SMS-MP inclusion on serum biochemical parameters of sika doe and suckling fawn are shown in Table 4. Compared to the control group of sika does, the serum concentrations of TP and GLO in the experimental group were significantly decreased (*p* < 0.05), the concentration of CHOL was greatly significantly decreased (*p* < 0.01), and the concentration of GLU was significantly increased (*p* < 0.05) with SMS-MP supplementation. In the results of the suckling fawns, it was found that the concentration of GLU was decreased, and the concentration of ALP was increased (*p* < 0.05) after the replacement of concentrate supplement with SMS-MP.

### 3.4. Effects on Serum Immune Globulin

According to the serum biochemical parameters of the tested sika deer (Table 5), there were no significant differences in immune globulin between the sika does and suckling fawns in the whole experiment process, regardless of whether SMS-MP existed (*p* > 0.05).

### 3.5. Effects on Plasma Amino Acids

Effects of SMS-MP on plasma amino acid of sika doe and suckling fawn are shown in Table 6. As the results showed, replacing partial concentrate supplements with SMS-MP significantly increased sika doe plasma Phe and Pro (*p* < 0.05), while it highly, significantly increased the level of Ala (*p* < 0.01). There were no differences in the plasma amino acid of suckling fawns by using of SMS-MP (*p* > 0.05).

## 4. Discussion

The SMS addition to ruminant feed has become increasingly accepted in recent years. However, the effects of feed SMS on sika doe and suckling fawn have not yet been reported. The present study was designed to explore the effects of adding SMS-MP to diets on the hematology of sika does and suckling fawns. Studies have shown that the replacement of 10% of the concentrate supplement with SMS-MP had no effect on hematology of sika does and suckling fawns. It can be concluded that SMS-MP can be used as a concentrate supplement for sika doe.

It was reported that SMS-MP can be used as ruminant feed without any deleterious effects on cattle eating behavior. Similarly, the current study found no effect of SMS-MP on feed intake and apparent digestibility of OM, EE, and CP in sika doe [20]. However, the SMS-MP reduced the digestibility of DM, which needs further study. Neutral detergent fiber (NDF) promotes digestion in ruminants, and dietary NDF has been reported to alter rumen fermentation and plasma metabolites [21]. In contrast, studies have shown that NDF has no effect on ruminant fermentability and digestibility in ruminants [22]. Therefore, the effect of SMS-MP on NDF digestibility for sika doe needs to be further confirmed.

The blood biochemical index is an important parameter for evaluating animal nutrition status [14]. TP contains ALB and α-, β-, and γ-GLO [23]; therefore, high concentrations of TP are associated with an increase in serum GLO concentration [24]. An elevated GLO concentration is probably the sign of chronic inflammation [25]. The GLO concentration of sika does decreased in the experimental group, indicating that SMS-MP could decrease the chronic inflammation caused by GLO. CHOL is an important component in the synthesis of bile acids and vitamin D in the liver and plays an important role in the formation of cell membranes in the body [26]. Studies have shown that low serum concentrations of CHOL could increase the risk of a retained placenta in perinatal dairy cows [27] and could cause fatty liver disease [28]. The results showed that SMS-MP could decrease the CHOL of sika doe, indicating that the reduction in CHOL may be beneficial to the calving of sika does. Carbohydrates in diets could be broken down into GLU and released in the systemic circulation to provide energy, and they played an important role in the synthesis of fatty acids [29]. The increased GLU concentration in the serum of SMS-MP-supplemented sika does suggests a lower rate of GLU used when SMS-MP was added in diets. However, the reason for the decreased GLU concentration of suckling fawns with supplemented SMS-MP in their diet is still unknown and needs further study. ALP is a very important enzyme that reflects the mineral metabolism in vivo [30]. The content of ALP increases physiologically during the growth period of suckling fawn bones, indicating that SMS-MP can promote the growth of suckling fawn bones. Serum BUN concentration is determined by the nitrogen level in the diet and can reflect the nitrogen metabolism of ruminants [31]. A high concentration of BUN in the serum indicates that the animal cannot fully utilize carbohydrates for energy. In addition, high levels of protein in the body may affect the immune system of cows, eventually leading to hyperammonemia [32]. There were no significant differences in BUN among the groups of the study. It can be suggested that SMS-MP as a concentrate supplement has no adverse effect on sika doe and suckling fawn.

Colostrum has a large number of immune-related components, mainly immunoglobulins. IgG, IgA, and IgM are three common immunoglobulins, which have the functions of preventing microbial invasion, regulating the homeostasis of the body’s immune environment, and controlling the proliferation of cancer cells [33]. It is well known that high-quality colostrum plays an important role in improving passive immunity and reducing the occurrence of diseases in newborn calves [34]. The concentrations of β-carotene and vitamin A in the colostrum of Japanese black cows were positively correlated with the concentrations of IgG and IgM [35]. During pregnancy, IgG is actively transported from the mother to the fetus through the placenta, thereby enhancing fetal immunity [36]. The predominant category of immunoglobulins in bovine colostrum and milk is IgG, while the largest proportion of antibodies in human milk is IgA [37]. Mucosal secretions contain a large amount of IgA, which prevents the invasion of pathogenic microorganisms by resisting the microorganisms on the mucosal surface [38]. By detecting the concentration of immunoglobulin in serum, the immune status of sika doe and suckling fawn can be observed indirectly. In our study, there were no significant differences in serum IgG, IgA, and IgM levels between the two groups, indicating that SMS-MP did not damage the immune system of the sika does and suckling fawns.

The blood of animals is the common medium that connects all organ systems in the body. Numerous studies have reported that various diseases can cause abnormal amino acid concentrations, and when specific organ systems are metabolically disordered, plasma amino acid concentrations can also be altered [39,40]. Ala is a non-essential amino acid formed by the transamination of pyruvate, which reflects the increased concentration of pyruvate under these conditions [41]. The current study showed that SMS-MP could increase the level of Ala in the plasma of sika doe, indicating that SMS-MP may improve the metabolism of nutrients in the body by increasing the content of pyruvate. As a special amino acid in collagen, Pro plays an important role in the structure of blood vessels. Deficiency of Pro may lead to impaired collagen synthesis and further damage to vascular structures, eventually leading to acute aortic dissection [42]. Phe is one of the essential amino acids in red blood cells and plasma [43]. The results indicated that SMS-MP could increase the levels of Phe and Pro, which may have some positive effects on the circulatory system of sika doe, while SMS-MP had no significant effect on the plasma amino acid content of suckling fawn.

SMS-MP can be consumed long-term as a concentrate supplement without harmful effects on the hematology of sika doe and suckling fawn. Although this study emphasized that SMS-MP had no effect on the digestion of sika doe, the digestion of suckling fawn requires further study.

## 5. Conclusions

In conclusion, replacing 10% of the concentrate supplement with SMS-MP had no effect on the digestion and feed intake of sika doe and did not change the hematology of sika doe and suckling fawn. The present findings confirm the applicability of SMS-MP as a supplement concentrate for sika doe. At the same time, using SMS, a waste resource, as a concentrate supplement for sika doe can not only reduce the breeding cost of sika deer, but also make full use of SMS to reduce environmental pollution.

## Figures and Tables

**Table 1 animals-12-01984-t001:** Ingredients of concentrate supplement (%).

Ingredients	Percentage
Corn	49.0
Soybean meal	49.0
NaCl	1.5
Stone powder	0.2
Bone meal	0.1
Ca(HCO_3_)_2_	0.2

**Table 2 animals-12-01984-t002:** Nutritional ingredients of concentrate supplement (%, DM).

Parameter	Control	Experimental
Crude protein (CP)	15.61 ± 0.21	15.33 ± 0.61
Energy (MJ/Kg)	16.64 ± 0.43	16.16 ± 0.26
Ether extract (EE)	12.72 ± 0.32	12.12 ± 0.28
Organic matter (OM)	95.72 ± 0.05	95.43 ± 0.05

**Table 3 animals-12-01984-t003:** Feed intake and nutrient digestibilities of sika doe.

Parameter	Control	Experimental
Organic matter intake (g·d^−1^)	1593.85 ± 43.58	1543.27 ± 120.54
Dry matter intake (g·d^−1^)	1665.67 ± 45.53	1617.12 ± 126.31
Digestibility of CP (%)	81.25 ± 3.85	81.15 ± 2.89
Digestibility of EE (%)	66.06 ± 7.00	67.65 ± 4.65
Digestibility of OM (%)	67.38 ± 6.21	67.03 ± 3.48
Digestibility of DM (%)	64.62 ± 6.08 ^a^	58.01 ± 8.50 ^b^

^a,b^ Different letters within the same row were significantly different (*p* < 0.05).

**Table 4 animals-12-01984-t004:** Comparison of serum biochemical parameters of sika doe and suckling fawn.

Parameter	Abbreviation	Sika Doe		Suckling Fawn	
		Control	Experimental	Control	Experimental
Albumin (g/L)	ALB	28.40 ± 0.40	28.17 ± 1.55	28.17 ± 0.76	28.47 ± 1.15
Total protein (g/L)	TP	67.70 ± 1.97 ^a^	63.23 ± 1.53 ^b^	52.93 ± 2.70	53.50 ± 2.88
Globulin (g/L)	GLO	39.30 ± 1.71 ^a^	35.07 ± 0.06 ^b^	24.77 ± 2.18	25.03 ± 2.27
Calcium (mmol/L)	Ca	2.43 ± 0.11	2.40 ± 0.14	2.38 ± 0.20	2.55 ± 0.08
Glucose (mmol/L)	GLU	6.46 ± 1.43 ^b^	9.63 ± 0.67 ^a^	8.55 ± 0.87 ^a^	6.82 ± 0.24 ^b^
Urea nitrogen (mmol/L)	BUN	4.96 ± 1.02	6.38 ± 0.48	5.48 ± 1.31	5.93 ± 0.49
Amylase (U/L)	AMY	10.00 ± 4.00	6.67 ± 3.21	7.67 ± 1.53	11.33 ± 2.31
Cholesterol (mmol/L)	CHOL	2.13 ± 0.11 ^A^	1.75 ± 0.04 ^B^	2.90 ± 0.27	2.23 ± 0.56
Alanine aminotransferase (U/L)	ALT	47.67 ± 6.11	50.33 ± 10.97	63.00 ± 21.00	72.0 ± 12.12
Total bilirubin (umol/L)	TBIL	4.64 ± 1.08	4.57 ± 1.59	4.10 ± 3.10	3.47 ± 0.85
Alkaline phosphatase (U/L)	ALP	199.67 ± 83.34	111.67 ± 15.95	391.67 ± 38.07 ^b^	541.67 ± 45.65 ^a^
Inosine (umol/L)	CRE	128.33 ± 38.55	168.00 ± 28.05	106.00 ± 13.75	111.67 ± 4.51
Creatine Kinase (U/L)	CK	170.33 ± 16.65	169.33 ± 13.61	397.00 ± 10.65	402.67 ± 9.02

^a,b^ Different letters within the same row were significantly different (*p* < 0.05). ^A,B^ Different letters within the same row were highly significantly different (*p* < 0.01).

**Table 5 animals-12-01984-t005:** Globulin of sika doe and suckling fawn (mg/mL).

Parameter	Sika Doe		Suckling Fawn	
	Control	Experimental	Control	Experimental
Immunoglobulin A (IgA)	62.60 ± 10.95	64.09 ± 9.67	81.53 ± 5.31	79.15 ± 4.36
Immunoglobulin G (IgG)	27.23 ± 13.75	36.34 ± 6.11	37.34 ± 20.98	25.00 ± 8.52
Immunoglobulin M (IgM)	3.08 ± 1.71	4.93 ± 1.24	3.29 ± 1.14	3.11 ± 0.75

**Table 6 animals-12-01984-t006:** Plasma amino acid of sika doe and suckling fawn (µmol/mL).

Parameter	Sika Doe		Suckling Fawn	
	Control	Experimental	Control	Experimental
Alanine (Ala)	17.12 ± 4.23 ^B^	28.77 ± 0.78 ^A^	15.10 ± 3.94	28.27 ± 9.07
Aspartic (Asp)	504.01 ± 46.53	521.34 ± 45.94	453.60 ± 37.73	479.07 ± 8.63
Isoleucine (Ile)	1283.28 ± 271.60	1397.45 ± 186.74	545.75 ± 43.28	597.55 ± 74.55
Leucine (Leu)	47.71 ± 3.92	54.23 ± 1.82	37.38 ± 7.66	48.65 ± 1.81
Phenylalanine (Phe)	59.62 ± 5.07 ^b^	69.23 ± 1.87 ^a^	51.36 ± 18.89	54.96 ± 0.88
Proline (Pro)	3.87 ± 0.31 ^b^	4.48 ± 0.05 ^a^	3.02 ± 0.94	3.44 ± 0.06
Serine (Ser)	4.79 ± 1.05	5.91 ± 0.01	3.21 ± 0.56	3.56 ± 0.04
Tyrosine (Tyr)	1.94 ± 0.68	2.43 ± 0.00	1.28 ± 0.20	1.16 ± 0.00
Valine (Val)	47.15 ± 8.28	53.05 ± 0.27	40.06 ± 4.27	37.66 ± 0.42

^a,b^ Different letters within the same row were significantly different (*p* < 0.05). ^A,B^ Different letters within the same row were very significantly different (*p* < 0.01).

## Data Availability

The dataset generated and/or analyzed during the current study is available from the corresponding author on reasonable request.

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
