# Peer review of "Hematological Changes in Sika Doe and Suckling Fawn Fed with Spent Mushroom Substrate of Pleurotus ostreatus"

_animals, 2022, doi:10.3390/ani12151984_

Round 1

Reviewer 1 Report

The work presented for review concerns the use of a 10% addition of Pleurotus ostreatus concentrate (SMS-MP) in order to reduce the cost of feeding sika deer and reduce environmental pollution with this mushroom substrate. The work presented for review concerns the use of a 10% addition of Pleurotus ostreatus concentrate (SMS-MP) in order to reduce the cost of feeding sika deer and reduce environmental pollution with this mushroom substrate.
The work was written correctly. I have no objections to it and propose to publish it in the journal Animals.

Author Response

Dear editor,

Thank you for your letter and for the reviewers’ comments concerning our manuscript entitled “Hematological Changes in Sika Doe and Suckling Fawn Fed with Spent Mushroom Substrate of Pleurotus ostreatus”. Those comments are all valuable and very helpful for revising and improving our paper, as well as the important guiding significance to our researches. We have studied the comments carefully and have made detailed correction and supplement which we hope meet with approval. We replied to all the reviewers' concerns point-by-point. The main corrections in the paper and the response to the reviewer’s comments are as following:

Comments from reviewers and Responses:

-Reviewer 1

Comments and Suggestions for Authors

The work presented for review concerns the use of a 10% addition of Pleurotus ostreatus concentrate (SMS-MP) in order to reduce the cost of feeding sika deer and reduce environmental pollution with this mushroom substrate. The work presented for review concerns the use of a 10% addition of Pleurotus ostreatus concentrate (SMS-MP) in order to reduce the cost of feeding sika deer and reduce environmental pollution with this mushroom substrate.
The work was written correctly. I have no objections to it and propose to publish it in the journal Animals.

Response: Thank you for supporting our research.

Reviewer 2 Report

1. Why the authors did not include the analysis and digestibility of fiber, considering at least the analysis of the NDF

2. The digestibility formula has already been included. But the reference to the analysis of acid-insoluble ash must be included.

3. I suggest including postprandial hours instead of meridian hours. Line 81

Author Response

Dear editor,

Thank you for your letter and for the reviewers’ comments concerning our manuscript entitled “Hematological Changes in Sika Doe and Suckling Fawn Fed with Spent Mushroom Substrate of Pleurotus ostreatus”. Those comments are all valuable and very helpful for revising and improving our paper, as well as the important guiding significance to our researches. We have studied the comments carefully and have made detailed correction and supplement which we hope meet with approval. We replied to all the reviewers' concerns point-by-point. The main corrections in the paper and the response to the reviewer’s comments are as following:

Comments from reviewers and Responses:

-Reviewer 2

Comments and Suggestions for Authors

  1. Why the authors did not include the analysis and digestibility of fiber, considering at least the analysis of the NDF

Response: Thanks for your advice. We considered the importance of NDF in the design of the trial. When we studied through the literature, we found that the effect of NDF on ruminants was not completely consistent, so we did not carry out related work, but we have supplemented the relevant information on NDF in detail in the manuscript Discussion as following:

“Neutral detergent fiber (NDF) promotes digestion in ruminants, dietary NDF have been reported to alter rumen fermentation and plasma metabolites [22]. In contrast, studies have shown that NDF has no effect on ruminant fermentability and digestibility in ruminants [23]. Therefore, the effect of SMS-MP on NDF digestibility of sika doe needs to be further confirmed.”

[22] Cao Y et al., Physically effective neutral detergent fiber improves chewing activity, rumen fermentation, plasma metabolites, and milk production in lactating dairy cows fed a high-concentrate diet. J Dairy Sci 2021;104:5631-5642.

[23] Kendall C, Leonardi C, Hoffman PC, Combs DK, Intake and milk production of cows fed diets that differed in dietary neutral detergent fiber and neutral detergent fiber digestibility. J Dairy Sci 2009;92:313-23.

  1. The digestibility formula has already been included. But the reference to the analysis of acid-insoluble ash must be included.

Response: Thanks for your advice. Reference related to acid-insoluble ash have been included in the manuscript as following:

“Nutrient digestibility: Using acid-insoluble ash as an internal digestibility marker, the apparent digestibility was calculated using the following equation [18]:”

[18]. Bao, K., et al., Effects of dietary manganese supplementation on nutrient digestibility and production performance in male sika deer (Cervus Nippon). Anim Sci J, 2017. 88(3): p. 463-467.

  1. I suggest including postprandial hours instead of meridian hours. Line 81

Response: Thanks for your advice. We have revised the issue in the manuscript as follows:

“The diets were restricted and offered 3 times per day at 4:30~5:30, 10:30~11:30, 16:30~17:30.”

Reviewer 3 Report

Reviewed manuscript "Hematological Changes in Sika Doe and Suckling Fawn Fed with Spent Mushroom Substrate of Pleurotus Ostreatus " (animals-1824701) just like the previous one about male Sika deer contains the results of very interesting research work of scientific and practical significance. These studies show two benefits of using spent mushroom substrate:

- feeding deer and reduce the production cost of velvet antler

- reducing waste at the same time.

Blood, on the other hand, is a very sensitive indicator of change, therefore it is a good marker.

The experiment was planned properly and carried out on sufficiently numerical material, female and fawns Sika deer kept on the professional farm.

Statistical analysis of the obtained results is correct.

Tables and figures presented the results and statistical data were constructed properly.

The discussion was carried out properly and the literature used in this part of the manuscript was chosen accordingly.

However, please improve:

-it would be good if the keywords were not repeated with the title of the manuscript

-line 63-65 - only the effect SMS on the antlers is described, it would be good to add that it can also affect the does and fawns

In summary - the manuscript contains valuable information, issues and after minor revision should be published in Animals.

Author Response

Dear editor,

Thank you for your letter and for the reviewers’ comments concerning our manuscript entitled “Hematological Changes in Sika Doe and Suckling Fawn Fed with Spent Mushroom Substrate of Pleurotus ostreatus”. Those comments are all valuable and very helpful for revising and improving our paper, as well as the important guiding significance to our researches. We have studied the comments carefully and have made detailed correction and supplement which we hope meet with approval. We replied to all the reviewers' concerns point-by-point. The main corrections in the paper and the response to the reviewer’s comments are as following:

Comments from reviewers and Responses:

-Reviewer 3

Comments and Suggestions for Authors

Reviewed manuscript "Hematological Changes in Sika Doe and Suckling Fawn Fed with Spent Mushroom Substrate of Pleurotus Ostreatus " (animals-1824701) just like the previous one about male Sika deer contains the results of very interesting research work of scientific and practical significance. These studies show two benefits of using spent mushroom substrate:

- feeding deer and reduce the production cost of velvet antler

- reducing waste at the same time.

Blood, on the other hand, is a very sensitive indicator of change, therefore it is a good marker.

The experiment was planned properly and carried out on sufficiently numerical material, female and fawns Sika deer kept on the professional farm.

Statistical analysis of the obtained results is correct.

Tables and figures presented the results and statistical data were constructed properly.

The discussion was carried out properly and the literature used in this part of the manuscript was chosen accordingly.

However, please improve:

-it would be good if the keywords were not repeated with the title of the manuscript

Response: Thanks for your advice. We have revised the keywords in the manuscript as shown below:

Keywords: sika deer; digestibility; blood hematological; spent mushroom substrate

-line 63-65 - only the effect SMS on the antlers is described, it would be good to add that it can also affect the does and fawns

Response: Thanks for your advice. The effect of SMS on does and fawns has been supplemented in the manuscript as follows:

However, As ruminants, there is no relevant study of SMS on the hematology of sika doe and suckling fawn. We speculate that sika deer can also digest SMS well and have no adverse effects on the hematology of sika doe and suckling fawn.

In summary - the manuscript contains valuable information, issues and after minor revision should be published in Animals.

Response: Thank you for supporting our research.
